# Medical honey for canine nasal intertrigo: A randomized, blinded, placebo-controlled, adaptive clinical trial to support antimicrobial stewardship in veterinary dermatology

Gabrielle Brosseau[1]*, Nadia Pagé[1], Caroline de Jaham[1], Jérôme R. E. del Castillo[2]

1 Department of Dermatology, Centre Vétérinaire DMV, Montreal, Quebec, Canada, 2 Quebec's Animal Pharmacology Research Group (GREPAQ), Department of Veterinary Biomedicine, University of Montreal, Saint-Hyacinthe, Quebec, Canada

* gbrosseau@centredmv.com

**Data Availability Statement:** All relevant data are within the paper and its Supporting Information files.

## Abstract

Intertrigo is a skin fold dermatitis often requiring recurrent treatment with topical antiseptics or antibiotics, which can select antimicrobial resistance. To minimize this risk, we tested the effectiveness of medical-grade Manuka honey at treating intertrigo as compared to a placebo hydrogel. We additionally characterized the culturable microbial flora of intertrigo and recorded any adverse effect with either treatment. During this randomized, placebo-controlled, double-blinded, adaptive group-sequential trial, the owners washed the affected sites on their dog with water, dried and applied a thin film of either the honey or the placebo product once daily for 21 days. Cytological and lesional composite scores, owner-assessed pruritus, and microbial cultures were assessed prior to treatment and on Day-22. The fixed effects of time, treatment, and animal-related variables on the pruritus and on each composite score, accounting for random dog effect, were estimated separately with generalized linear mixed models for repeated count outcomes (α = 0.05). The null hypothesis of equal treatment effects was rejected at the first interim analysis. The placebo (n = 16 dogs) outperformed the medical honey (n = 13 dogs) at improving both the cytological score (Treatment×Time = -0.35±0.17; P = 0.04) and clinical score (Treatment×Time = -0.28±0.13; P = 0.04). A microbial burden score higher than 4 increased the severity of the cytological score (dichotomous score: 0.29±0.11; P = 0.01), which in turn increased the severity of the clinical score and pruritus score. For every unit increase in cytological score, the linear predictor of clinical score increased by 0.042±0.019 (P = 0.03), and the one of pruritus score increased by 0.12±0.05 (P = 0.01). However, medical honey outperformed the placebo at alleviating the dog's owner-assessed pruritus after statistically controlling for masking effects (Time = -0.94±0.24; P = 0.002; and Treatment×Time = 0.80±0.36; P = 0.04). Unilateral tests of the least-square mean estimates revealed that honey only significantly improved the pruritus (Hommel-adjusted P = 0.003), while the placebo only improved the cytological and clinical scores (Hommel-adjusted P = 0.01 and 0.002, respectively). Taken together, these results question the value of Manuka honey at treating nasal intertrigo in dogs.

**Funding:** The study was supported by the Canadian Academy of Veterinary Dermatology Research Grant (https://www.cavd.ca/) attributed to GB and the Centre Vétérinaire DMV (https://centredmv.com/en/). The funders had no role in study design, data collection and analysis, decision to publish, or preparation of the manuscript.

**Competing interests:** The authors have declared that no competing interests exist.

## Introduction

For the past two decades, bacterial resistance to antibiotics has emerged as a global crisis that jeopardizes our ability to treat infectious diseases in both humans and animals [1]. Being increasingly part of the family, household dogs share a significant part of their skin microbiota with their owners [2], and can transfer potentially zoonotic bacteria such as *Staphylococcus pseudintermedius* to both their owners [3] and the veterinary personnel [4]. In addition, dogs can disseminate such pathogens throughout the domestic and clinical environments [5, 6], in which canine and human bacteria may exchange their antimicrobial resistance genes [7].

Intertrigo (skin fold dermatitis) is a frictional dermatitis that occurs in areas where two skin surfaces are intimately apposed, in which poor ventilation, accumulation of body secretions and debris favour the development of secondary superficial infection [8]. The condition is more prevalent in brachycephalic dogs like pugs, English and French bulldogs [9–11], in dogs with specific lip, tail or perineal anatomical conformations [12, 13], and in obese dogs [14]. Unless the underlying cause is either controlled or permanently corrected, intertrigo will recur and require periodical treatment with antibiotics or antiseptics [8].

Antimicrobial use in animals exposes their microbiota to antimicrobial resistance selective pressure [15]. It is urgent that companion-animal veterinarians adopt antimicrobial steward-ship strategies to preserve their efficacy and availability [16]. This involves a 5R approach (i.e., responsibility, reduction, replacement, refinement, and review) for avoiding any unwarranted use of antimicrobials [17], and improving their necessary use by means of an optimization feedback loop that considers all available diagnostic-therapeutic process and outcomes data, along with judicious use guidelines. A recent clinical trial in a companion animal hospital illustrates the potential of this approach: within two years of implementation, the prevalence of methicillin- resistant *Staphylococcus intermedius* bacterial group recovered from their patients decreased by 70% [18].

Three expert panels issued antibiotic use guidelines for the treatment of skin infections in small animals [19–22]: the latest emphasized the use of topical antibiotics or antiseptics as sole antibacterial treatment for surface or superficial pyoderma. Several topical formulations and active ingredients are available in companion animal medicine, but high quality, randomized controlled trials evaluating their efficacy against skin infections are scarce [23]. Besides, resistance to some of their active ingredients (e.g. mupirocin, fusidic acid, and chlorhexidine) is emerging in human medicine in parallel to their expanding topical use [24, 25]. Finding efficacious alternative topical products for treating intertrigo may save these antimicrobials for more severe skin infections requiring their use.

Medical-grade honey, developed under standard conditions and sterilized by gamma irradiation [26], is a promising alternative for the topical therapy of intertrigo and other surface/superficial cutaneous infections in companion animals [23]. Owing to a variety of antimicrobial compounds, medical honey is active in vitro against Gram-positive and negative bacteria, including drug-resistant isolates and bacterial biofilms [26, 27]. Slightly decreased, yet rapidly reversible bacterial sensitivity appears under prolonged exposure to Manuka honey [28]. In addition, it promotes wound cicatrization and angiogenesis [29], and may provide anti-inflammatory effects that reduce oedema and maceration [30]. In dogs, the clinical efficacy against surface pyoderma of L-Mesitran® Ointment (Triticum medical, Maastricht, Limbourg, Netherlands), which contains 48% Yucatan honey, did not significantly differ from that of a 3% chlorhexidine shampoo [31]. However, this result is inconclusive evidence for the therapeutic potential of all medical honeys because of inadequate statistical testing (41% of lesions were found on the same 7 dogs), and because L-Mesitran® additionally contains 7 other wound-healing ingredients [32–38]. Lacking a honey-free ointment arm and an inert

treatment arm, this trial result may have benefited from unaccounted antimicrobial potentiation and/or placebo effects [39].

The purpose of this study was to investigate the use of a 100% pure medical-grade honey (Medihoney[TM]) as an alternative to topical antimicrobials in the control of canine nasal intertrigo. Because of its active substances, its therapeutic efficacy in human intertrigo [40], and the similarities between the canine and human forms of this disorder, we hypothesize that medical-grade Manuka honey would be safe and clinically superior to a placebo topical therapy at treating nasal intertrigo in brachycephalic dogs. The main objective of this study was (1) to compare the severity of intertrigo clinical signs and cytological findings before and after a 21-days treatment course with either Medihoney[TM] or a honey-scented placebo hydrogel. Our secondary objectives were (2) to assess how each treatment affected the culturable microbial flora of nasal intertrigo, which is currently undefined, and (3) to record any adverse effect with either treatment.

## Materials and methods

### Trial design and bioethical approval

This study was a randomized, placebo-controlled, double-blinded, adaptive group sequential trial. The blinded interim statistical analyses of accumulating data allowed the following trial amendments:

i. Early stop the trial because:

 a. At least 3 consecutive dogs in a given treatment group had adverse reactions with no other concomitant event than treatment onset or,

 b. The estimated difference in therapeutic effect size was either:

 i. Large enough to reject the null hypothesis $\mu1 = \mu2$ (i.e. efficacy),

 ii. Small enough to accept the null hypothesis (i.e. futility),

ii. Else, reassess the sampling sizes and continue the trial.

The University of Montreal Animal Bioethics Committee approved the study protocol and owner informed consent form (approval # 18-Rech-1939).

### Animals and inclusion/exclusion criteria

Between March 2018 and February 2019, we recruited incoming cases diagnosed with nasal intertrigo at Centre Vétérinaire DMV, irrespective of breed, sex or age, and recruited additional cases through advertisement on social media groups of brachycephalic dog breeds. Before entering the study, dog owners received the study objectives, interventions and protocol, and signed the informed consent form (S1 Appendix). Participating owners could withdraw their dog from the trial at any time and for any reason. We applied the following exclusion criteria on eligible dogs:

i. Within 14 days of enrolment:

 a. Administration of systemic antibiotic or antifungal drugs,

 b. Topical application of antibiotic, anti-inflammatory, antiseptic or anti-fungal drugs on the affected nasal fold,

 c. Intra-ocular topical application of antibiotic or anti-inflammatory drugs (due to potential tear accumulation on the affected site),

 d. Changes in chronic medications (whether dermatological or not).

ii. Within 8 weeks of enrolment:

 a. Administration of extended-release or long-lasting glucocorticoids,

 b. Dietary change in dogs with suspected food allergy.

iii. Within 12 weeks of enrolment:

 a. Changes in allergen-specific immunotherapy regimen.

## Test articles

Gamma-irradiated MediHoney$^{TM}$ 100% antibacterial medical honey, lot #1840 (Integra Life-Sciences, Princeton, New Jersey, United States) were purchased from Cardinal Health Canada (Vaughan, Ontario, Canada) and sent unopened to Gentès et Bolduc, Pharmacists (St-Hyacinthe, Quebec, Canada) for aseptic transfer into screw capped, opaque, sealable plastic tubes with a capacity of 15 g. In parallel, they compounded a placebo hydrogel made of starch, glycerol and a pluronic lecithin mixture, supplemented with 1% v/w of ethyl phenylacetate 98% purity (Toronto Research Chemicals, Toronto, Ontario, Canada) to provide a honey-like fragrance. They transferred the placebo hydrogel into plastic tubes identical to the ones filled with honey, and labeled them with the random treatment code letters A or B. The antimicrobial inertness of the placebo hydrogel was confirmed by verifying the absence of growth inhibition halos around 1 μL and 10 μL hydrogel drops poured on a Mueller-Hinton agar inoculated with a $1.5 \times 10^8$ cfu/mL saline suspension of a *S. pseudintermedius* clinical isolate, and aerobically incubated for 18 hours at 35°C.

## Treatments, monitoring, randomization, and post-study disease management

On Day-1 and 22, each dog received a general health and complete dermatological examinations, including the clinical and cytological assessments of nasal intertrigo, and swabbing of the affected area for microbial culture. An animal health technician randomized each dog's treatment with a coin toss and kept the random coding undisclosed to both the owners and investigators until the end of the final statistical data analysis.

Once daily from Day-1 to 21, dog owners gently cleaned the affected skin fold with a clean, water-moistened towel or tissue, sponged it with a dry towel or tissue, and delicately applied the given product on the affected skin surface with a clean finger to obtain a thin, uniform layer. Owners informed the investigators of any adverse event as soon as detected and avoided using other topical treatments or cleaning products on the affected site during the trial period. The investigators contacted the dog owners weekly to ensure treatment compliance and verified reporting accuracy on Day-22.

After trial completion, dogs with inadequate improvement (i.e. composite clinical score > 2) shifted to a topical chlorhexidine treatment made of Baxedin® 20% (Omega Laboratories Ltd, Montreal, Quebec, Canada) diluted 1:10 in sterile water, which was applied identically to the previous treatment. These cases were re-examined on Day-42 and refractory cases shifted to topical antibiotic therapy, according to the results of Day-22 antimicrobial sensitivity testing, until clinical resolution (S1 Dataset).

## Data collection

**Clinical lesion assessment.** On Day-1 and 22, the first investigator (GB) photographed the lesion site and estimated its clinical severity with the Canine Atopic Dermatitis Extent and Severity Index, version 4 (CADESI-4), which grades each of erythema, lichenification, and the combination of excoriation/alopecia as follows [41]: 0 = none; 1 = slight; 2 = moderate; 3 = severe and extensive. We graded identically the abundance of exudate at lesion site. A second investigator (NP or CdJ) examined the photographs to confirm the grading of all four items (S2 Dataset). Finally, we added the four grades to obtain the composite clinical severity score (from 0 to 12) at Day-1 and Day-22.

**Cytological assessment.** On Day-1 and 22, the first investigator (GB) spread a specimen of material swabbed from the lesion site on a microscopy slide, dyed it with a modified Wright stain, and examined ten microscopic fields at 400x magnification as described elsewhere [42]. The abundances of cocci, rods, yeasts, and inflammatory cells were graded each as follows [43]: 0 = none seen; 1 = occasionally present but slide must be scanned carefully for detection; 2 = present in low numbers, but detectable rapidly without difficulties; 3 = present in larger numbers and detectable rapidly without any difficulties; 4 = massive amount present. A second investigator (NP or CdJ) examined all slides to confirm the grading given to each item (S2 Dataset). Afterwards, we added the four grades to obtain the composite cytological severity score (from 0 to 16) at Day-1 and Day-22.

**Microbiological assessment.** IDEXX Reference Laboratories (Markham, Ontario, Canada) suspended the swabbed material of each animal in peptone broth, inoculated a standardized volume of suspension on a Mueller-Hinton plate using the 4-quadrant streak method, and incubated the agars for 18 hours at 35˚C without $CO_2$ supplementation. Then, they graded the abundance of each distinct colony type as follows: 0 = no growth; 1 = growth only in the first quadrant with less than 10 colonies; 2 = growth only in the first quadrant with 10 colonies or more; 3 = growth up to the second quadrant; 4 = growth beyond the second quadrant. Afterwards, isolated colonies of each type were identified (S2 Dataset), and the antimicrobial sensitivity of pathogenic isolates tested per manufacturing instructions with the Vitek® 2 automated platform (S1 Table). MALDI-TOF techniques differentiated the isolates belonging to the *Staphylococcus intermedius* group [44], and methicillin-resistant *S. pseudintermedius* (MRSP) detected with the oxacillin disk diffusion method [45]. Finally, we added the grades of each cultivable microorganism to obtain composite microbial abundance scores at Day-1 and Day-22.

**Pruritus assessment.** On Day-1 and 22, owners quantified the perceived pruritus intensity of their dogs with a visual analog scale (VAS) published elsewhere [46], to which 10 evenly spaced intervals were drawn aside (S2 Dataset).

## Statistical analyses

**Blinded interim analyses.** We designed our trial with the SEQDESIGN procedure (SAS 9.4 version, SAS Institute Inc., Cary, NC, U.S.A.) using the following settings:

i. The pivotal variable comparing the efficacies of the tested products was the Treatment×Time interaction in a repeated-measures, generalized linear mixed model of Poisson-distributed composite clinical scores, with Treatment (A or B) as second fixed factor, Time of assessment as a fixed covariate, and a random intercept for each dog.

ii. The null and alternative hypotheses respectively were $H_0$: μ(A)—μ(B) = 0, and $H_1$: absolute value of μ(A)—μ(B) = 0.693, corresponding to one product producing on average a 2-unit

larger variation in composite clinical score than the other, accounting for their respective Day-1 values.

iii. The trial had up to three analysis stages: two interim and the final analysis. As all analyses were blinded, two-sided hypothesis testing was performed at α = 0.05 type-I error, and β = 0.10 type-II error probabilities. The standard deviation of [μ(A)—μ(B)] was assumed equal to 0.97, which yields 95% confidence limits of 1.4 and 2.8 score units for the estimated difference at the first interim analysis.

iv. At each stage of the trial, the O'Brien-Fleming α- and β-boundaries respectively determined the rejection or acceptance of H$_0$ [47].

Table 1 presents the resulting sample sizes and α- and β-boundaries.

### Final inferential analysis

Once the trial stopped, we performed exploratory data analyses on the distributions of breed, age, sex, reproductive status, and of the carriage of microbial isolates across treatment groups. A bilateral t-test for two independent samples with equal variances compared the age of dogs, and bilateral Fisher exact tests compared the levels of the other variables. If one or more tests rejected the null hypothesis of equal means or proportions, we controlled the false discovery rate with the Benjamini-Hochberg procedure for independent or positively dependent statistics [48]. We additionally examined the bivariate relationships among the outcome variables and their relationships with the animal characteristics, stratified by treatment group, to identify potential predictors for refining our statistical models.

Then, we built generalized linear mixed models for repeated Poisson-distributed outcomes, identical to the one used for the interim analysis of clinical scores, for analyzing the compound microbial abundance, cytological, and pruritus scores. We refined all four models by including, when appropriate, the age, breed, sex, or the compound scores as additional predictors to control potential confounding in the variation of outcome variables.

All analyses with repeated measures, generalized linear mixed models for Poisson-distributed outcomes, were performed with SAS, 9.4 version (SAS Institute Inc., Cary, NC, USA), using the Laplace approximation for maximum likelihood estimation, and "sandwich" estimator of the parameter covariance matrices. The Akaike information criterion and the distribution of conditional Pearson residuals determined the model that best fitted the data.

Once the analyses completed and blind coding revealed, we added the final models with two one-sided a priori tests among least-square mean estimates for determining which treatment improved the tested outcome variable (i.e. H$_0$: μ(i) ≥ 0, where i = placebo or honey). We

**Table 1. Calculated sample sizes per trial stage, expected Fisher information, and standardized Z values of the two-sided O'Brien-Fleming bounds for rejection or acceptance of the null hypothesis of equal mean effects.**

| Trial stage | Sample size | Fisher Information gathered | | Rejection statistical bounds | | Acceptation statistical bounds | |
|---|---|---|---|---|---|---|---|
| | | Proportion | Expected | Lower | Upper | Lower | Upper |
| Interim #1 | 28.9 | 0.33 | 7.67 | -3.41 | 3.41 | | |
| Interim #2 | 57.7 | 0.67 | 15.3 | -2.41 | 2.41 | -1.06 | 1.06 |
| Final | 86.6 | 1.00 | 23.0 | -1.97 | 1.97 | -1.97 | 1.97 |

This table presents the expected increases in Fisher information, i.e., the reciprocal of variance, of the estimated Treatment×Time effect on the linear predictor of composite clinical score, as a function of the sample size at each trial stage (first interim, second interim and final statistical analysis). In addition, the table presents the statistical boundaries for the rejection or acceptance of the null hypothesis of equal mean effects at each stage. The higher the value of the Fisher information is, the higher the precision of the estimated effect size.

controlled the familywise type-I error rate for this series of tests by subjecting their raw P-values to the step-up Bonferroni adjustment of Hommel [49].

## Results

All but the last section of the results were completed before disclosure of the following blinding code: Treatment A = Placebo; Treatment B = Honey.

### Sample sizes and blinded interim analyses

The first stage of the trial recruited 35 dogs, 19 of which received the placebo and other 16 received the medical honey. Of these, 30 dogs completed the trial (17 placebo and 13 honey), but the data of one dog given placebo was discarded because its chronic allergy medication (oclacitinib; Apoquel$^{TM}$, Zoetis Canada Inc., Kirkland, Quebec, Canada) was terminated during the trial, in breach of trial's exclusion criteria. Therefore, we performed the interim statistical analysis with the data of 16 dogs treated with placebo and 13 with honey.

Table 2 reveals that, after collecting 33.5% of the maximum sample size (i.e., 29/86.6; Table 1), the first interim analysis gathered sufficient Fisher information to reject the null hypothesis of equal mean effects of the tested articles: Z = -2.16, outside the -1.97 lower statistical boundary. The estimated effect of Treatment×Time on the linear predictor of clinical score was -0.37 with 95% confidence interval = [-0.71, -0.04]: the negative values indicate that the placebo decreased the clinical score more effectively than honey (P = 0.03).

Of the five dogs not completing the study, one per treatment group was withdrawn because of adverse reactions to the applied product (increased itching or redness of the treated area), and the other three (two placebos and one honey) were lost to follow-up for lack of motivation of the owners.

### Blinded exploratory data analysis

Dogs completing the trial ranged from 8 months to 12 years of age, most (25/29) were neutered and 19/29 were males. Most dogs were Pugs (14/29) or English Bulldogs (13/29), the remainder being French Bulldogs. The microorganisms most frequently isolated from intertrigo lesions were methicillin-sensitive *Staphylococcus pseudintermedius* (in 19/29 dogs), *Streptococcus canis* (15/29), *Escherichia coli* and *Pseudomonas aeruginosa* (both in 7/29 dogs). The Day-1 culture of case #1 (placebo) yielded a MRSP isolate additionally resistant to the potentiated sulfonamides, macrolides, fluoroquinolones, tetracyclines, and gentamicin, but sensitive to amikacin.

None of the tested variables significantly differed among treatment groups at Day-1 (S2 Table).

**Table 2. Result of the first blinded interim analysis to accept or reject the trial's null hypothesis.**

| Trial stage | Sample size | Fisher information gathered | | Rejection / acceptation statistical bounds | Z-Test of Trt×Time | |
|---|---|---|---|---|---|---|
| | | Proportion | Actual | | Estimate | Action |
| Interim #1 | 29 | 1.00 | 34.2 | ±1.97 | -2.16 | Reject $H_0$ |

This table presents the sample size, actual Fisher information value and result of the blinded first interim statistical analysis testing the null hypothesis of equal Treatment×Time means on the linear predictor of the intertrigo composite clinical score, i.e., H0: μ(A)—μ(B) = 0. As a recall, we hypothesized that medical-grade Manuka honey would be safe and clinically superior to placebo at treating nasal intertrigo. The expected amount of Fisher information that the trial would gather for testing the null hypothesis of equal treatment effects was 23.0 for a maximum sampling size of 86.6 dogs (Table 1).

## Microbial growth diversity and abundance

The microbial cultures yielded at least one isolate in all but one dog at Day-1, with medians of 3 isolates on Day-1 and 2 isolates on Day 22 in placebo-treated dogs, and of 2 isolates at both times in honey-treated dogs. *Proteus mirabilis*, *Streptococcus canis* and *Pseudomonas aeruginosa* showed the largest, yet modest variations with respect to Day-1 prevalence values (S2 Table): +2, -4 and -3 of 29 dogs, respectively. The Day-22 prevalence of each microorganism did not significantly differ among treatment groups after controlling for the false discovery rate (S3 Table).

The composite microbial abundance scores were multimodal in both treatment groups, and both showed a temporal trend of regression towards the mean, i.e. extreme scores at Day-1 tended to converge to mid-value scores by Day-22 (Fig 1). None of Treatment, Time, and Treatment×Time significantly influenced the composite microbial abundance score (Table 3; $P > 0.05$), and none of the recorded patient characteristics improved the model's goodness of fit to the data. The abundance grades of isolated microorganisms are available in the S2 Dataset.

## Composite cytological score

Table 4 presents the estimated fixed effects of Treatment, Time, and Treatment×Time on the composite cytology score. The model's goodness of fit to the data significantly improved by

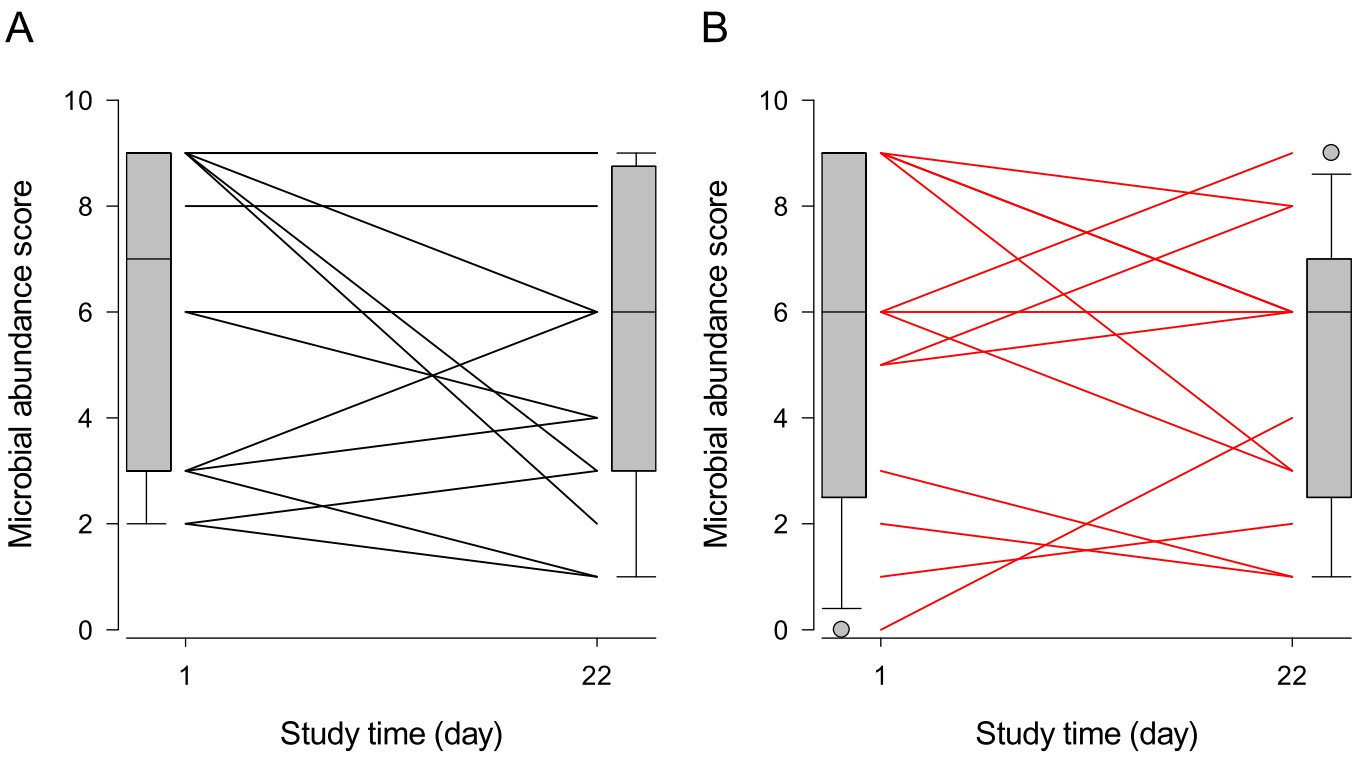

**Fig 1. Individual initial and final microbial abundance scores of dogs treated with topical placebo or honey.** Legend: Panel A = placebo, panel B = honey. The composite microbial abundance scores were obtained for each dog at Day-1 and 22 by adding the growth scores (i.e. from 0 = no growth, to 4 = visible growth beyond the 2nd quadrant) of all lesion-site organisms cultured on Mueller-Hinton agar plates using the 4-quadrant streaking method. Each line connects the Day-1 and Day-22 scores for one or more dogs sharing the same initial and final scores; boxplots at the margins of the graph illustrate the empiric distributions of the Day-1 and Day-22 scores. The tips of the bottom and top whiskers of the boxplots represent the 5th and 95th percentiles, the bottom and top of the gray-shaded box represent the 25th and 75th percentiles, and the horizontal line inside the box represents the median of the empirical distribution. The gray filled circles represent the individual values outside the 5th– 95th percentiles of the distribution.

**Table 3. Estimated fixed effects of Treatment, Time, and Treatment×Time on the linear predictor of the composite microbial abundance score.**

| Effect | Level | Est. | SE | 95% C.I. Bounds | | Type-3 Test | | |
|---|---|---|---|---|---|---|---|---|
| | | | | Lower | Upper | Den. DF | F-Statistic | P-value |
| Intercept | | 1.57 | 0.18 | 1.19 | 1.94 | | | |
| Time | 22 | -0.09 | 0.17 | -0.43 | 0.26 | 28 | 0.89 | 0.35 |
| Trt | A | 0.15 | 0.22 | -0.31 | 0.61 | 28 | 0.44 | 0.51 |
| Trt×Time | A-22 | -0.03 | 0.21 | -0.45 | 0.40 | 28 | 0.02 | 0.90 |

95% C.I., 95% confidence interval; > 4, composite microbial score value greater than 4; A, Placebo group; A 22, Placebo group at Day-22; d22, Day-22; Den., denominator; DF, degrees of freedom; Est., estimated parameter value; SE, standard error; Trt, treatment; Trt×Time, Treatment×Time interaction. Note: the numerator DF = 1 for all type-3 tests. This table presents the estimated coefficients of the linear predictors of Poisson-distributed composite microbial abundance scores, standard errors, boundaries of the 95% confidence intervals, and results of the type-3 statistical testing of fixed effects. The composite microbial abundance scores were obtained for each dog at Day-1 and 22 by adding the growth scores (i.e. from 0 = no growth, to 4 = visible growth beyond the 2nd quadrant) of all lesion-site organisms cultured on Mueller-Hinton agar plates using the 4-quadrant streaking method.

adding a dichotomous version (0 = score ≤ 4; 1 = score > 4) of the composite microbial score (Table 4), whose threshold was set based on the visual inspection of Fig 1.

The Day-1 cytological score values were higher for placebo than for honey-treated dogs, a significant difference (Trt: P = 0.003), and microbial abundance scores greater than 4 significantly increased the cytological scores in both treatment groups (Mi_Bin: P = 0.01). At variance with honey-treated dogs, whose cytological scores remained unchanged after trial completion (Time: P = 0.21), placebo-treated dogs had significantly lower scores with respect to Day-1 levels (Trt×Time: P = 0.04).

Fig 2 depicts the effect of each treatment on the composite cytological score of dogs: most placebo-treated dogs had negative slopes, whereas the honey-treated dogs varied randomly.

## Clinical lesion score

Table 5 presents the results of statistical testing of composite clinical lesion scores. We refined the model used in the interim analysis and improved its goodness of fit by introducing the composite cytological score as an additional fixed factor and replacing the dog's random intercept with its dichotomized composite microbial abundance scores.

**Table 4. Estimated fixed effects of Treatment, Time, Treatment×Time, and dichotomous microbial burden score on the linear predictor of the composite cytology score.**

| Effect | Level | Est. | SE | 95% C.I. Bounds | | Type-3 Test | | |
|---|---|---|---|---|---|---|---|---|
| | | | | Lower | Upper | Den. DF | F-Statistic | P-value |
| Intercept | | 1.27 | 0.14 | 0.98 | 1.57 | 32 | | |
| Time | d22 | 0.07 | 0.14 | -0.22 | 0.35 | 26 | 1.67 | 0.21 |
| Trt | A | 0.64 | 0.16 | 0.32 | 0.95 | 32 | 10.3 | 0.003 |
| Trt×Time | A-22 | -0.35 | 0.17 | -0.69 | -0.01 | 26 | 4.47 | 0.04 |
| Mi_Bin | > 4 | 0.29 | 0.11 | 0.07 | 0.51 | 26 | 7.07 | 0.01 |

95% C.I., 95% confidence interval; > 4, composite microbial score value greater than 4; A, Placebo group; A 22, Placebo group A at Day-22; d22, Day-22; Den., denominator; DF, degrees of freedom; Est, estimated parameter value; Mi_Bin, dichotomized microbial abundance score (i.e., 0 = score ≤ 4, 1 = score > 4); SE, standard error; Trt, treatment; Trt×Time, Treatment×Time interaction. Note: The numerator DF = 1 for all type-3 tests. This table presents the estimated coefficients of the linear predictors of Poisson-distributed composite cytological scores, standard errors, boundaries of the 95% confidence intervals, and results of the type-3 statistical testing of fixed effects for the placebo group. The composite cytological scores were calculated for each dog on Day-1 and 22 by adding the abundance scores (0 = none, to 4 = massive amounts) of cocci, rods, yeasts and inflammatory cells.

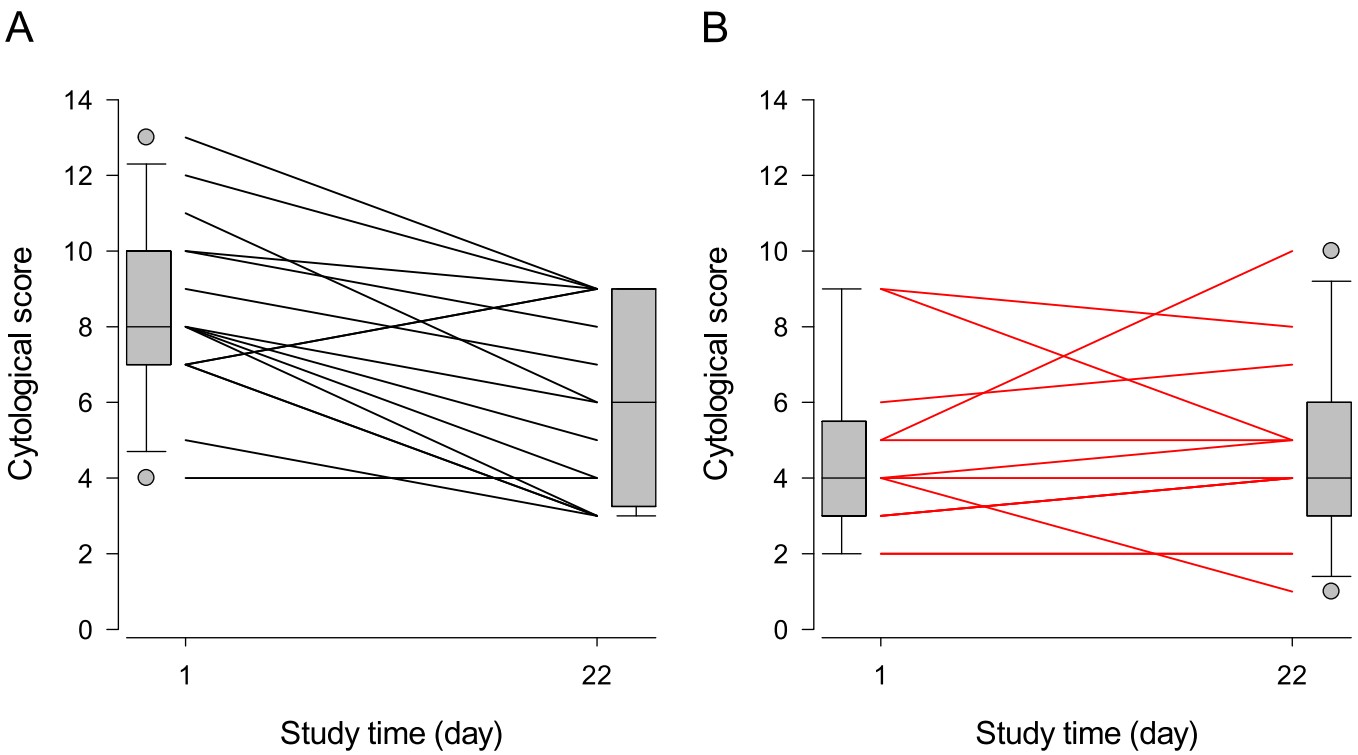

**Fig 2. Individual initial and final composite cytological scores of dogs treated with topical placebo or honey.** Legend: Panel A = placebo, panel B = honey. The composite cytological scores were calculated for each dog on Day-1 and 22 by adding the abundance scores (0 = none, to 4 = massive amounts) of cocci, rods, yeasts and inflammatory cells. Each line connects the Day-1 and Day-22 scores for one or more dogs sharing the same initial and final scores; boxplots at the margins of the graph illustrate the empiric distributions of the Day-1 and Day-22 scores. The tips of the bottom and top whiskers of the boxplots represent the 5th and 95th percentiles, the bottom and top of the gray-shaded box represent the 25th and 75th percentiles, and the horizontal line inside the box represents the median of the empirical distribution. The gray filled circles represent the individual values outside the 5th– 95th percentiles of the distribution.

According to this refined model, the Day-1 score values did not significantly differ among treatment groups (Trt: P = 0.32), and the scores decreased by the end of the trial in both the placebo (Treatment×Time: P = 0.04) and honey groups (Time: P = 0.004). Moreover, the severity of cytological score increased the severity of clinical score in both groups (Cy_Score: P = 0.03).

**Table 5. Estimated fixed effects of Treatment, Time, Treatment×Time, and cytological score on the linear predictor of the composite clinical score.**

| Effect | Level | Est. | SE | 95% C.I. Bounds | | Type-3 Test | | |
|---|---|---|---|---|---|---|---|---|
| | | | | Lower | Upper | Den. DF | F-Statistic | P-value |
| Intercept | | 1.41 | 0.12 | 1.17 | 1.64 | | | |
| Time | d22 | -0.04 | 0.09 | -0.22 | 0.14 | 36 | 9.68 | 0.004 |
| Trt | A | 0.28 | 0.14 | -0.02 | 0.57 | 36 | 1.03 | 0.32 |
| Trt×Time | A-22 | -0.28 | 0.13 | -0.55 | -0.01 | 36 | 4.42 | 0.04 |
| Cy_Score | | 0.042 | 0.019 | 0.003 | 0.080 | 36 | 4.86 | 0.03 |

95% C.I., 95% confidence interval; A, Placebo; A 22, Placebo at Day-22; d22, Day-22; Den., denominator; DF, degrees of freedom; Cy_Score, composite cytological score; Est, estimated parameter value; SE, standard error; Trt, treatment; Trt×Time, Treatment×Time interaction. Note: The numerator DF = 1 for all type-3 tests. This table presents the estimated coefficients of the linear predictor of the Poisson-distributed composite clinical score of intertrigo lesions, standard errors, boundaries of the 95% confidence intervals, and results of type-3 statistical testing. The composite clinical score was calculated by adding the scores (0 = none, to 3 = severe and extensive) of erythema, lichenification, the combination of excoriation/alopecia, and exudate present at lesion site.

Fig 3 depicts the effects of treatment on the composite clinical scores during the trial. At variance with honey, placebo decreased the clinical scores of most treated dogs.

### Owner-perceived pruritus of dogs

Table 6 presents the results of statistical testing of the integer values of the pruritus VAS. The exploratory data analysis revealed that Pugs had lower and more homogeneous scores than English and French Bulldogs. In addition, the pruritus score correlated with the composite cytological score, and showed clusters similarly to the composite clinical scores. Therefore, the model included the fixed effects of Treatment, Time, Treatment×Time, and composite cytological score, and the random clinical scores of dogs, with separate covariances for Pugs and Bulldogs.

According to this model, the Day-1 pruritus score did not significantly differ among treatment groups (Trt: $P = 0.13$), honey significantly decreased the pruritus scores (Day: $P = 0.002$), and placebo showed significantly less anti-pruritic activity than honey (Treatment×Time: $P = 0.04$). Finally, the cytological score increased the perceived intensity of dog's pruritus for all treatment groups and assessment times ($P = 0.02$).

Fig 4 depicts the effects of treatment on the integer of visual analog scale values of owner-perceived pruritus in dogs: although honey significantly decreased the perceived intensity of pruritus more effectively than placebo (Table 6), this difference was unapparent in the graphs.

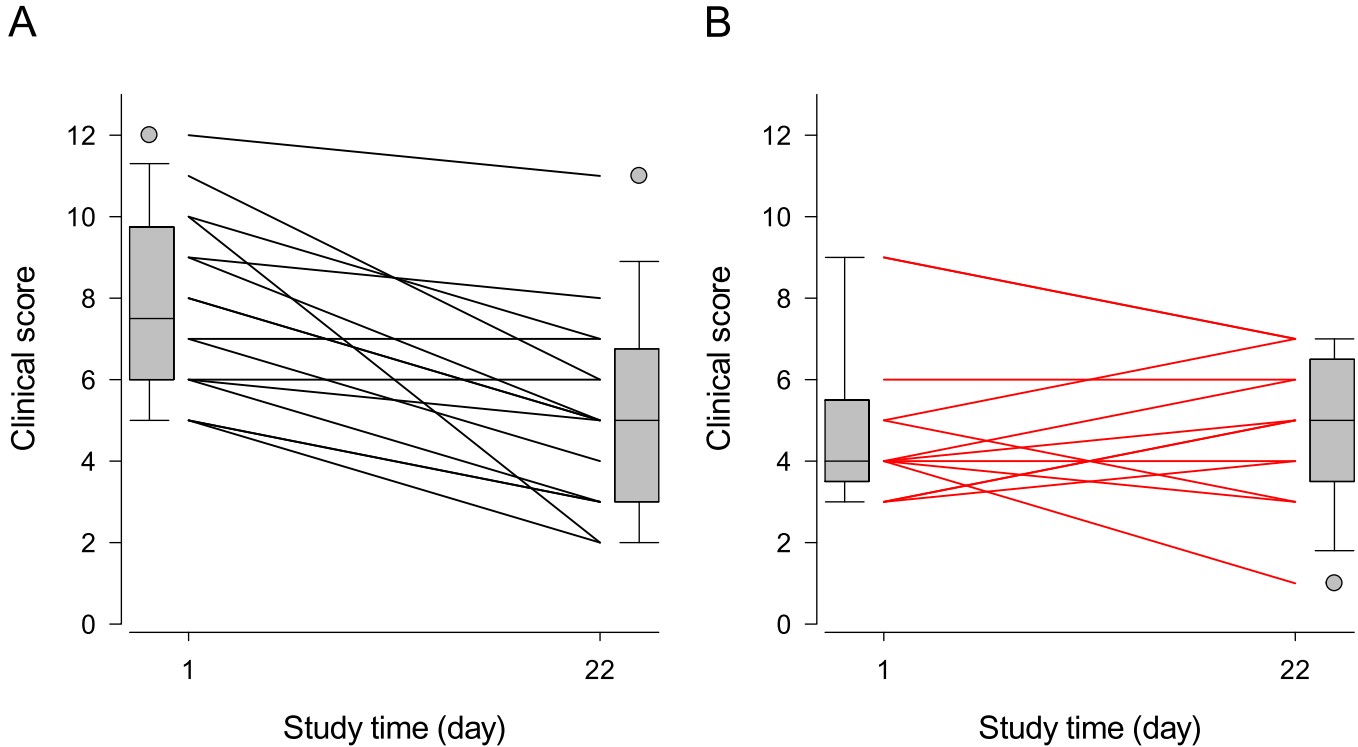

**Fig 3. Individual initial and final composite clinical scores of dogs treated with topical placebo or honey.** Legend: Panel A = placebo, panel B = honey. The composite clinical score was calculated by adding the scores (0 = none, to 3 = severe and extensive) of erythema, lichenification, the combination of excoriation/alopecia, and exudate present at lesion site. Each line connects the Day-1 and Day-22 scores for one or more dogs sharing the same initial and final scores; boxplots at the margins of the graph illustrate the empiric distributions of the Day-1 and Day-22 scores. The tips of the bottom and top whiskers of the boxplots represent the 5[th] and 95[th] percentiles, the bottom and top of the gray-shaded box represent the 25[th] and 75[th] percentiles, and the horizontal line inside the box represents the median of the empirical distribution. The gray filled circles represent the individual values outside the 5[th]– 95[th] percentiles of the distribution.

**Table 6. Estimated fixed effects of Treatment, Time, Treatment×Time, and cytological score on the linear predictor of the integer values of the pruritus VAS.**

| Effect | Level | Est. | SE | 95% C.I. Bounds | | Type-3 Test | | |
| --- | --- | --- | --- | --- | --- | --- | --- | --- |
| | | | | Lower | Upper | Den. DF | F-Statistic | P-value |
| Intercept | | -0.06 | 0.40 | -0.89 | 0.77 | | | |
| Time | d22 | -0.94 | 0.24 | -1.45 | -0.44 | 24 | 12.31 | 0.002 |
| Trt | A | 0.25 | 0.47 | -0.72 | 1.22 | 24 | 2.52 | 0.13 |
| Trt×Time | A-22 | 0.80 | 0.36 | 0.06 | 1.54 | 24 | 5.00 | 0.04 |
| Cy_Score | | 0.12 | 0.05 | 0.03 | 0.22 | 24 | 6.91 | 0.01 |

95% C.I., 95% confidence interval; A, Placebo; A-22, Placebo at Day-22; d22, Day-22; Den., denominator; DF, degrees of freedom; Cy_Score, composite cytological score; Est, estimated parameter value; SE, standard error; Trt, Treatment; Trt×Time, Treatment×Time interaction. Note: The numerator DF = 1 for all type-3 tests. This table presents the estimated coefficients of the linear predictor of Poisson-distributed integers of the visual analog scale of owner-perceived pruritus of dogs, standard errors, boundaries of the 95% confidence intervals, and type-3 statistical testing of the fixed effects. The scores are the integer of the VAS values of owner-perceived pruritus of their dogs.

Noteworthy, the cytological score, clinical score and breed are hidden determinants of the time-course of pruritus VAS illustrated in Fig 4. An appraisal of their effects on the significance of the fixed effects of the model is available in S2 Appendix.

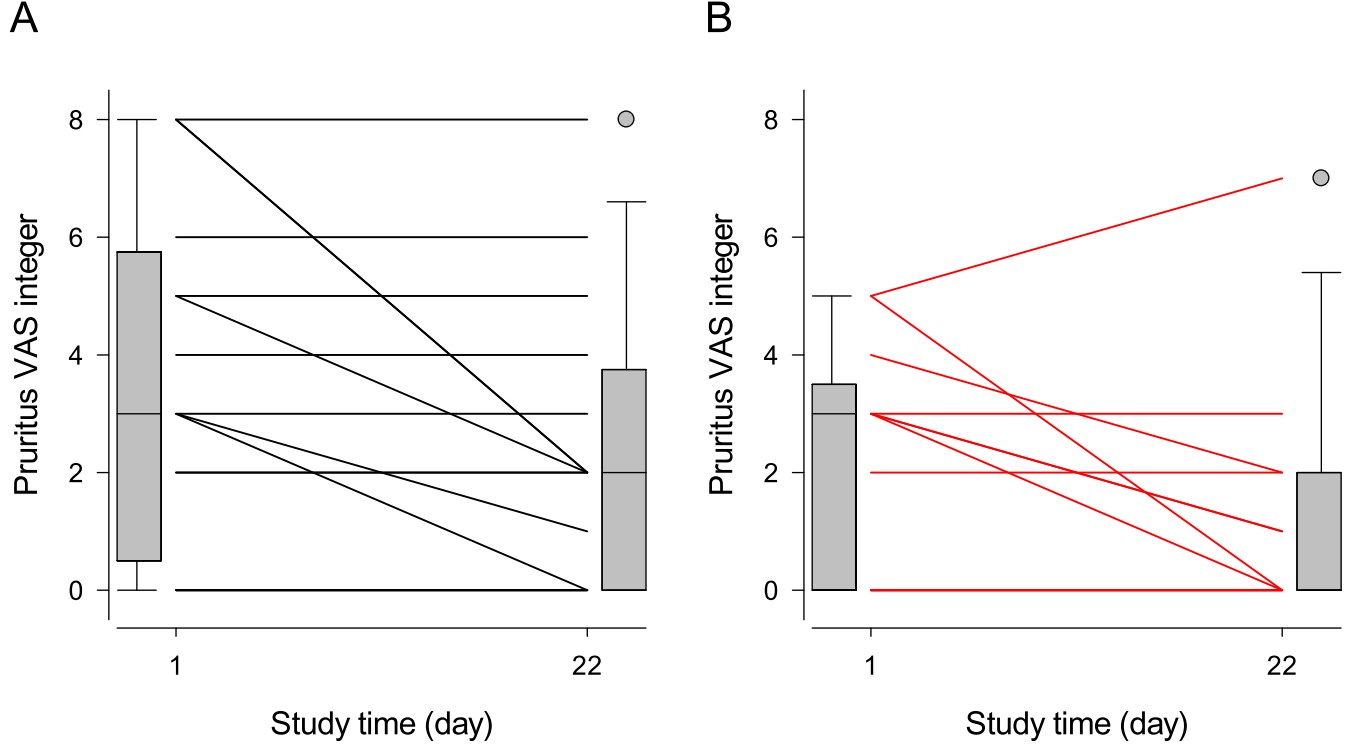

**Fig 4. Individual initial and final pruritus VAS scores of dogs treated with topical placebo or honey.** Legend: Panel A = placebo, panel B = honey. The scores are the integer of the VAS values of owner-perceived pruritus of their dogs. Each line connects the Day-1 and Day-22 scores for one or more dogs sharing the same initial and final scores; boxplots at the margins of the graph illustrate the empiric distributions of the Day-1 and Day-22 scores. The tips of the bottom and top whiskers of the boxplots represent the 5th and 95th percentiles, the bottom and top of the gray-shaded box represent the 25th and 75th percentiles, and the horizontal line inside the box represents the median of the empirical distribution. The gray filled circles represent the individual values outside the 5th– 95th percentiles of the distribution.

**Table 7. Unilateral statistical testing of the efficacies of the placebo and medical honey on the tested therapeutic outcomes.**

| Treatment | Outcome | Est. | SE | DF | t-Statistic | Raw P-val. | Hommel Adj.-P |
|---|---|---|---|---|---|---|---|
| **Placebo** | **Mi_score** | -0.06 | 0.06 | 28 | -0.91 | 0.19 | 0.48 |
| | **Cy_score** | -0.14 | 0.05 | 26 | -3.12 | 0.002 | 0.01 |
| | **Cl_score** | -0.17 | 0.04 | 26 | -3.90 | <0.001 | 0.002 |
| | **Pruritus** | -0.07 | 0.11 | 24 | -0.62 | 0.27 | 0.54 |
| **Honey** | **Mi_score** | -0.04 | 0.09 | 28 | -0.49 | 0.31 | 0.63 |
| | **Cy_score** | 0.03 | 0.07 | 26 | 0.48 | 0.68 | 0.68 |
| | **Cl_score** | -0.02 | 0.04 | 26 | -0.48 | 0.32 | 0.63 |
| | **Pruritus** | -0.47 | 0.12 | 24 | -3.85 | <0.001 | 0.003 |

Adj.-P, adjusted probability value; DF, degrees of freedom; Cl_score, composite clinical lesion score; Cy_score, composite cytological score; Est, estimated parameter value; Mi_score, composite microbial abundance score; SE, standard error. Note. This table presents the least-square mean estimates, standard errors, and results of unilateral statistical testing of the efficacy of the placebo and medical honey topical products at mitigating the tested therapeutic outcomes, with raw and Hommel-adjusted probability values, which hold the familywise type-I error rate at $\alpha = 0.05$ for the combined eight tests.

## Hommel-adjusted statistical testing of the therapeutic benefit of tested articles

Unilateral statistical testing (i.e., targeting a score reduction) of the therapeutic efficacies of the placebo and medical honey are summarized in Table 7, and representative patients illustrated in Fig 5. Only placebo achieved significant cytological and clinical benefits (Cy_score:

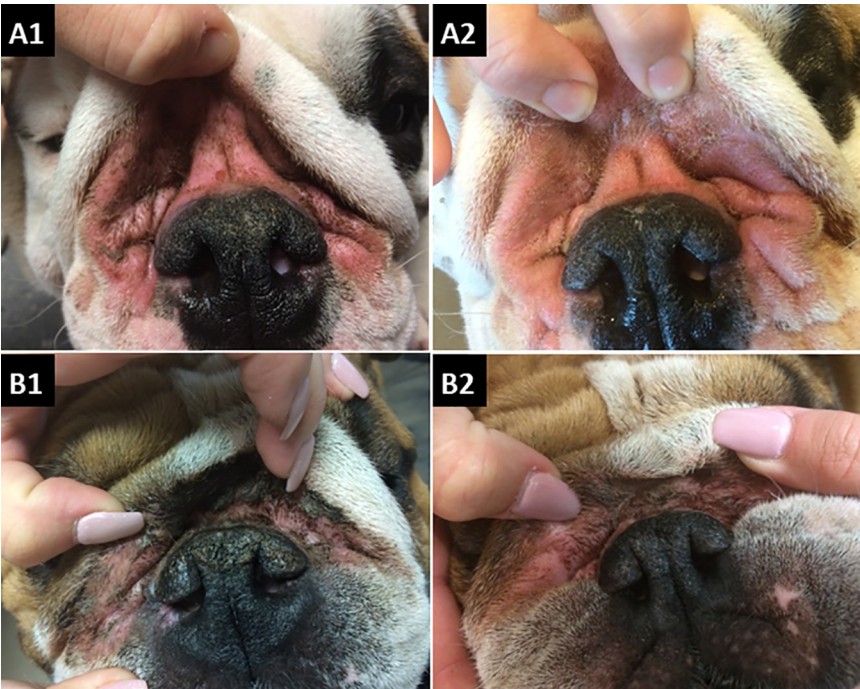

**Fig 5. Nasal intertrigo lesions of typical dogs before and after treatment with placebo or honey.** Legend: A1, dog #11 prior to treatment with placebo; A2, dog #11 after 21 days of treatment with placebo; B1 dog #9 prior to treatment with medical-grade honey; B2, dog #9 after 21 days of treatment with medical-grade honey.

P = 0.01, Cl_score: P = 0.002), and only honey significantly alleviated the pruritus VAS (Pruritus: P = 0.003).

## Discussion

The purpose of this study was to investigate the use of a medical-grade 100% Manuka honey (Medihoney™) in the control of canine nasal intertrigo. We hypothesized that honey would be safe and clinically more effective at controlling this disorder than a placebo. Since no prior information could assist us at estimating the effect sizes and associated variances of placebo and honey in treating canine intertrigo, we circumvented the problem of arbitrarily fixing sample sizes by using an adaptive group-sequential design [50].

At variance with our expectations, honey did not outperform the placebo hydrogel. The only variable for which the honey prevailed was the owners' perceived pruritus intensity of their dog (Tables 6 and 7). In contrast, the cytological and clinical lesion composite scores only improved significantly (i.e. decreased) with placebo (Table 7). The composite microbial abundance scores tended to regress towards the mean with either treatment, with variations independent of trial design or patient characteristics (Table 3).

We excluded one dog for missing a week of oral oclacitinib during the trial. This chronic medication stabilized its allergic pruritus for years at inclusion time. The dog presented on Day-22 a flare of itching and erythema on multiple body areas, including the nasal fold, a clinical portrait typical of the rebound phenomenon caused by the abrupt withdrawal of oclacitinib [51].

The significant anti-pruritic effect of honey in our dogs (Table 6) mirrors the results of a clinical trial where the same product was tested in human patients with the same skin disorder [40]. However, Fig 4 failed to provide visual confirmation of this finding, because the plotted data had no correction for the masking effects of breed, cytological score and clinical score tackled by the statistical model. We identified these effect modifiers during the exploratory data analysis, confirming the critical importance of this initial stage of statistical analysis (S2 Appendix). Study designs using stiffer inclusion criteria or stratified randomization that limit the variation or distribute more evenly the dog breeds and their initial microbial, cytological and clinical scores across study arms prior to treatment onset, may better determine whether this antipruritic effect not only has statistical significance but also clinical significance. However, these alternative designs imply additional patient visits, lower patient inclusion rates, and a delayed treatment onset with the test articles that potentially hinders the well-being and disease remission of patients. The canine anti-pruritic compounds in honey may include its flavonoids chrysin and galangin [52]. In vivo murine and in vitro models both reveal that chrysin curbs the vascular inflammation by decreasing the endothelial permeability and NF-κB mediated recruitment of inflammatory cells [53], while galangin inhibits the infiltration of mast cells and the release of IFN-γ, IL-4, IL-5, IL-13, IL-31 and IL-32 by the keratinocytes [54]. Of these, IL-31 is a chief pruritogenic agent in canine atopic dermatitis [55], but its role in intertrigo remains undefined.

In addition to the above, our finding that the cytological score significantly increased the pruritus score is compatible with the recent demonstration that IL-31, histamine, and other pruritogenic substances can generate the itch sensation in dogs just as in other animals [56]. This verification has been elusive in dogs, as previous studies using intradermal injections reported visually distinct, yet statistically non-significant, dose-response relationships for the number and duration of itching episodes [57]. However, failure to reject the null hypothesis likely resulted from type-2 error, because discrete counts and time-to-event outcomes do not comply with the distributional assumptions of repeated-measures ANOVA [58]. By using a

generalized linear mixed model for count outcomes, our cytological score reliably estimated the abundance of inflammatory cells and microorganisms at lesion site, which in turn contributed to the release of the chemical determinants of our pruritus score.

At variance with our placebo, honey was ineffective at improving the cytological and clinical components of canine nasal intertrigo. This result was surprising as numerous studies report the in vitro antimicrobial activity of Manuka honey, our placebo was free of known antimicrobial or wound-healing ingredients, and we tested its antimicrobial inertness using a representative target pathogen. All ingredients used in manufacturing placebo were either 98% pure or pharmaceutical grade, product compounding and labeling followed ISO 9001–2008 standards, and both placebo and honey produced their expected effects on the pruritus score. In addition, we weekly monitored the compliance of participating owners to treatment procedures and verified that the dog did not lick the product from the treated site. Moreover, we searched the presence on the cytology smears of *Simonsiella* spp. [59], a highly prevalent canine oral bacterium whose size under the microscope is unmistakably larger relative to skin bacteria. Therefore, we believe that the support for either an accidental inversion of the random blinding codes or an insufficient treatment contact time is weak.

Alternatively, it is conceivable that these disappointing results of honey relate to its high methylglyoxal content [60]. Methylglyoxal, the main antimicrobial agent of Manuka honey [61], additionally is an electrophilic reactive intermediate product of several enzymatic pathways that produce free radicals [62]. When methylglyoxal concentrations exceed the detoxifying capacity of the cytosolic glycoxalases and catalytic glutathione, cells suffer an oxidative stress that depletes their pool of glutathione, yields methylglyoxal adducts on their proteins and DNA, and the accumulation of advanced glycosylation end-products furthers toxicity [63]. The risks of adverse effects of exogenous methylglyoxal exposure in patients chronically exposed to hyperglycemia is a topic of concern [64]: twelve of our 29 dogs were overweight (score > 5 on a 9-level scale; 75th percentile = 6, max. = 9), but body condition correlated with none of our microbial, cytological, clinical and pruritus scores.

To the best of our knowledge, this is the first study describing the culturable microflora of nasal intertrigo. Our most prevalent microorganism was MSSP, a skin commensal whose predominance as a pathogen is consistent with other types of superficial skin infections [8, 65]. Interestingly MRSP was only isolated on Day-1 from two placebo-treated dogs. The other most frequently isolated agents were *S. canis*, *E. coli* and *P. aeruginosa*. *S. canis* is a commensal resident of canine nasal passages [8]. Isolation of *E. coli* is uncommon in canine pyoderma, but microbiome techniques have detected *Enterobacteriaceae* on the dog's dorsal nose [66]. The ubiquitous *P. aeruginosa* was an expected finding that is present on the dorsal nose and other body parts of the dog [66]. A previous microbial culture study of canine lip fold intertrigo reported the isolation of *S. pseudintermedius*, *E. coli*, *Malassezia spp.*, *Simonsiella spp.* and *Pseudomonas spp.* [12]: all were isolated in our study except *Simonsella spp.*, a normal oral inhabitant of dogs [59]. The other microorganisms were isolated rarely from our treated lesions, making their prevalence estimates unreliable.

The microbial burden increased the severity of cytological scores, which in turn increased the severity of clinical and pruritus scores, showing that the microbial burden is a crucial and active component of canine nasal intertrigo. The majority of cases, irrespective of treatment, were polymicrobial but culture-independent sequencing methods would assuredly detect additional microorganisms. We could not have access to microbiota characterization techniques for this study, but the impact of the microbial burden on the severity of intertrigo demonstrated in our study warrants further research to establish the relationships between microbiota composition and the pathogenesis of the disease [67].

To the best of our knowledge, ours is the first study to use formally an adaptive trial design methodology with live patients in a private veterinary hospital, and the second one to use it in veterinary medicine [68]. In the earlier study, a drop-the-loser adaptive design tested hypotheses pertaining the risk of fire ignition during laser surgery using a cadaveric rodent model [68]: at the interim data analysis, a Haybittle-Peto boundary determined which mask type had significantly higher incidence of fire, and the remaining cadavers tested only the winner mask. Previous animal studies claiming the use of an adaptive trial design most likely were classical dose-escalation trials, because neither calculated the statistical boundaries for testing a null hypothesis at each interim analysis [69, 70], and adaptive decisions following the interim analysis were not prespecified [71].

Our study findings have several limitations. First, the number of dogs included was the minimum required to reject our null hypothesis of equal treatment effects on the clinical score. Differences in cytological and pruritus scores additionally were significant, but the estimated difference in microbial burden score would require 1395 dogs/group to reach significance at $\alpha$ = 0.05 and $\beta$ = 0.10. Large-sample clinical studies may further our findings by identifying covariates that modulate the efficacy of topical medical honey at controlling other skin infections. Second, we performed this trial in a specialized dermatology service of a large veterinary referral center, but believe that our results also apply to first-line small animal hospitals because all but one dog were admitted without referral for a first episode of canine nasal intertrigo. Third, a high proportion of recruited patients had a history of allergies, which could influence the lesion scores. The environment and nutrition of the dogs was not standardized, and it is unknown if it could have influenced the results. However, this aspect has the advantage of better mimicking the true environmental situation of dogs. Fourth, the application technique was identical for both products and explained identically to all dog owners, but we cannot rule out possible inter-owner variation in the actual method of application. It is as well possible that the owners under-reported a lack of compliance. Fifth, we had no access to analytical techniques for confirming the purity of the tested lot of medical-grade honey [72, 73], but product end-users likewise rely on the reputation and internal quality control procedures of the product manufacturer and distributors. Sixth, since the most prevalent canine skin pathogens are sensitive to Manuka honey in vitro [74–76], and sub-inhibitory in vitro exposure to this product increases the tolerance, viability and biofilm production of *P. multocida* [27], the lack of antimicrobial effect recorded during this trial may suggest that the tested honey was applied scantly or too sparsely. However, the dog owners received instruction to apply the product per manufacturer's instructions, and the repeated cleaning, drying and product reapplication may both irritate the treated zone and decrease owner compliance. To finish, our clinical lesion scores derives from the CADESI-4 scale, which validation for grading the severity of intertrigo is pending.

## Conclusions

Based on the results presented in Tables 3–5 and 7 and Figs 1–3, we cannot recommend the use of medical grade Manuka honey as an alternative to topical antimicrobials in the control of canine nasal intertrigo, as its therapeutic efficacy was inferior to our placebo hydrogel. The alleviation of pruritus (Table 6 and Fig 4) seems a mild, short-term gain for the well-being of both the dog and its owner, but its inability to resolve the microbial, cytological and clinical components of the disease may outweigh this benefit. Our adaptive trial design methodology opens new grounds for testing the efficacy of veterinary therapies.

## Supporting information

**S1 Appendix. Owner informed consent form.**
(DOCX)

**S2 Appendix. Development of the statistical model of owner perceived pruritus VAS.** (PDF)

**S1 Dataset. Clinical outcome of the tested articles, and for further treatments with 2% Chlorhexidine or topical antibiotics.** A, placebo; Age_yr, years of age; Age_mo, additional months of age; B, honey; EngBD, English bulldog; F, female; FraBD, French bulldog; I, sexually intact; ID_num, identification number; M, male; N, neutered; Tr1, initial treatment (during trial); Tr2, second treatment (post-trial); Tr3, third treatment (post-trial). (XLSX)

**S2 Dataset. Patient characteristics, isolated microorganisms, and cytological, clinical, and microbial grading.** A, placebo; Age_yr, years of age; Age_mo, additional months of age; B, Honey; Cocci_M, cytological abundance of cocci (mentor grading); Cocci_R, cytological abundance of cocci (1st author grading); E_coli, *Escherichia coli*; E_faecalis, *Enterococcus faecalis*; EngBD, English bulldog; Eryth_M, severity of erythema (mentor grading); Eryth_R severity of erythema (1st author grading); Excor_M, severity of excoriations or alopecia (mentor grading); Exsud_M, abundance of exudate (mentor grading); Excor_R severity of excoriations or alopecia (1st author grading); Exsud_R, abundance of exudate (1st author grading); F, female; FraBD, French bulldog; Hafnia, *Hafnia* species; I, sexually intact; ID_num, identification number; Infcell_M, cytological abundance of inflammatory cells (mentor grading); Infcell_R, cytological abundance of inflammatory cells (1st author grading); K_vari, *Klebsiella variicola*; K_sp, *Klebsiella* species; Lecler, *Leclercia adecarboxylata;* Lichen_M, severity of lichenification (mentor grading); Lichen_R, severity of lichenification (1st author grading); M, male; M_pachy, *Malassezia pachydermatis*; MRSP, methicillin-resistant *Staphylococcus pseudintermedius*; MSSP, methicillin-sensitive *Staphylococcus pseudintermedius*; N, neutered; Norm_-flora, normal bacterial flora; P_aeru, *Pseudomonas aeruginosa*; P_mirab, *Proteus mirabilis*; Rods_M, cytological abundance of rods (mentor grading); Rods_R, cytological abundance of rods (1st author grading); Sta_aureus, *Staphylococcus aureus*; Sta_schl, *Stahpylococcus schleferii*; Str_canis, *Streptococcus canis*; VAS, owner-perceived intensity of pruritus; Weiss, *Weissella confusa*; Yeasts_M, cytological abundance of yeasts (mentor grading); Yeasts_R, cytological abundance of yeasts (1st author grading). (XLSX)

**S1 Table. Distributions of antimicrobial sensitivity status for the four most prevalent bacterial isolates.** E.coli, *Escherichia coli*; I, intermediate; MRSP, methicillin-resistant *Staphylococcus pseudintermedius*; MSSP, methicillin-sensitive *Staphylococcus pseudintermedius*; P. aeruginosa, *Pseudomonas aeruginosa*; R, resistant; S, sensitive; S.canis, *Streptococcus canis*. (XLSX)

**S2 Table. Mean (standard deviation) age, distributions of breed, sex, and sterilization status of dogs recruited for the clinical trial, number of dogs carrying culturable microorganisms on their nasal intertrigo lesions before treatment with the tested topical products, and results of statistical comparisons of treatment groups with bilateral Student or Fisher exact tests.** Count, cell frequency in a two-way table; Fisher, bilateral Fisher exact test; Mean, arithmetic mean; SD, standard deviation; Student, bilateral Student t-test for two independent samples with identical variances; Trt, treatment. Count variables: number of dogs with matching data (e.g. Pugs vs. French or English Bulldogs, neutered vs. sexually intact dogs, carriers vs. non-carriers of a given microbial isolate at the site of intertrigo). Abbreviations of microbial isolates: E.coli, *Escherichia coli*; E.faecalis, *Enterococcus faecalis*; Hafnia sp., *Hafnia* species; Klebsiella sp., *Klebsiella* species; K.variicola, *Klebsiella variicola*; L.adecarb., *Leclercia*

*adecarboxylata*; M.pachyderm., *Malassezia pachydermatis*; MRSP, methicillin-resistant *Staphylococcus pseudintermedius*; MSSP, methicillin-sensitive *Staphylococcus pseudintermedius*; P. aeruginosa, *Pseudomonas aeruginosa*; P.mirabilis, *Proteus mirabilis*; S.aureus, *Staphylococcus aureus*; S.canis, *Streptococcus canis*; S.schleferii, *Stahpylococcus schleferii*; W.confusa, *Weissella confusa*.
(DOCX)

**S3 Table. Number of dogs carrying distinct microbial isolates on their nasal intertrigo lesions after 21 consecutive days of topical treatment with placebo or Manuka medical honey, and results of multiple statistical testing with bilateral Fisher exact tests before and after application of the Benjamini-Hochberg adjustment for controlling the false discovery rate.** BH-adj., Benjamini-Hochberg adjustment for controlling the false discovery rate; Raw, uncorrected probability value; Trt, treatment. Abbreviations of microbial isolates: E.coli, *Escherichia coli*; E.faecalis, *Enterococcus faecalis*; Hafnia sp., *Hafnia* species; K.variicola, *Klebsiella variicola*; Klebsiella sp., *Klebsiella* species; L.adecarb., *Leclercia adecarboxylata*; M.pachyderm., *Malassezia pachydermatis*; MRSP, methicillin-resistant *Staphylococcus pseudintermedius*; MSSP, methicillin-sensitive *Staphylococcus pseudintermedius*; P.aeruginosa, *Pseudomonas aeruginosa*; P.mirabilis, *Proteus mirabilis*; S.aureus, *Staphylococcus aureus*; S. canis, *Streptococcus canis*; S.schleferii, *Stahpylococcus schleferii*; W.confusa, *Weissella confusa*.
(DOCX)

## Acknowledgments

The authors thank the participating dog owners, Benjamin Tanguay, Pharm.D. (Gentès & Bolduc Pharmacists, St-Hyacinthe QC) and Hani Dick, Ph.D. (IDEXX Reference Laboratory, Brampton ON) for sharing their expertise, and Valérie Fleury-Gravel, TSA, for the randomization of cases and explanations to the dog owners.

## Author Contributions

**Conceptualization:** Gabrielle Brosseau, Nadia Pagé, Caroline de Jaham, Jérôme R. E. del Castillo.

**Data curation:** Gabrielle Brosseau, Jérôme R. E. del Castillo.

**Formal analysis:** Jérôme R. E. del Castillo.

**Funding acquisition:** Gabrielle Brosseau, Caroline de Jaham.

**Investigation:** Gabrielle Brosseau, Nadia Pagé, Caroline de Jaham.

**Methodology:** Jérôme R. E. del Castillo.

**Project administration:** Gabrielle Brosseau, Jérôme R. E. del Castillo.

**Resources:** Gabrielle Brosseau, Caroline de Jaham.

**Software:** Jérôme R. E. del Castillo.

**Supervision:** Nadia Pagé, Caroline de Jaham.

**Validation:** Jérôme R. E. del Castillo.

**Visualization:** Gabrielle Brosseau, Jérôme R. E. del Castillo.

**Writing – original draft:** Gabrielle Brosseau, Jérôme R. E. del Castillo.

**Writing – review & editing:** Gabrielle Brosseau, Nadia Pagé, Caroline de Jaham, Jérôme R. E. del Castillo.

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
