## [Decision Letter · Decision Letter 0]

26 Mar 2020

PONE-D-20-01412

Medical honey for canine nasal intertrigo: a randomized, blinded, placebo-controlled, adaptive clinical trial to support antimicrobial stewardship in veterinary dermatology

PLOS ONE

Dear Dr Brosseau,

Thank you for submitting your manuscript to PLOS ONE. After careful consideration, we feel that it has merit but does not fully meet PLOS ONE’s publication criteria as it currently stands. Therefore, we invite you to submit a revised version of the manuscript that addresses the points raised during the review process.

Both reviewers agree that the study has been well done and the results are interesting, and also that the limitations have been addressed appropriately. Both note that it could be improved by some shortening, and in particular that the tables could be presented with less detail that would make the take-home message easier to see. Please also note the comment from Reviewer #1 that it is possible that sub-therapeutic levels of honey could enhance bacterial growth if the MIC for the infecting bacteria is not known - it would be helpful to assess bacterial susceptibility if possible.

Please also address the other comments raised by the reviewers.

We would appreciate receiving your revised manuscript by May 10 2020 11:59PM. To enhance the reproducibility of your results, we recommend that if applicable you deposit your laboratory protocols in protocols.io, where a protocol can be assigned its own identifier (DOI) such that it can be cited independently in the future. For instructions see: http://journals.plos.org/plosone/s/submission-guidelines#loc-laboratory-protocols

We look forward to receiving your revised manuscript.

Kind regards,

Dee A. Carter, PhD

Academic Editor

PLOS ONE

Journal Requirements:

2. At this time, we request that you  please report additional details in your Methods section about the 'MediHoney' product used in this study. Please include any chemical characterization or measures of purity performed on this product. Please also report any additional processing steps taken other than irradiation, such as whether the honey was heated at any point.

4. Please upload a copy of Figure 5, to which you refer in your text on page 24. If the figure is no longer to be included as part of the submission please remove all reference to it within the text.

Reviewers' comments:

Reviewer's Responses to Questions

**Comments to the Author**

1. Is the manuscript technically sound, and do the data support the conclusions?

Reviewer #1: Yes

Reviewer #2: Yes

2. Has the statistical analysis been performed appropriately and rigorously? 

Reviewer #1: Yes

Reviewer #2: Yes

3. Have the authors made all data underlying the findings in their manuscript fully available?

Reviewer #1: Yes

Reviewer #2: Yes

4. Is the manuscript presented in an intelligible fashion and written in standard English?

Reviewer #1: Yes

Reviewer #2: Yes

5. Review Comments to the Author

Reviewer #1: An interesting study comparing the use of medical honey and a placebo hydrogel for the treatment of canine intertrigo. My major comments to the authors are below:

1. Tables in the main text are difficult to follow and main conclusions are not clear. I am not convinced that all of the statistical information included in the tables are wholly valuable (perhaps better suited for supplementary), and find that they detract from the overall conclusions provided in the text from the data in the tables. It would be of benefit to simplify these tables and include in the legend what each of the testing scores are measuring/how they were determined.

2. Is the sensitivity of the bacterial isolates/species identified in the study to honey known / can they be determined? It could be of value to determine the concentration of honey required to inhibit the growth of these species (e.g. MIC testing) as sub-inhibitory concentrations can often boost the growth of some bacteria (and other microbes) - likely due to an abundance of carbon sources (sugars) in honey. This has also been noted previously for bacteria growing in biofilms (single and mixed species), and the growth of the bacterial species in this study could also be in biofilm form.

3. The authors have identified the limitations of the study thoroughly. In line with my comment above, if the bacteria tested are not sensitive to honey, the dose (concentration/amount) of honey applied in the study may not be sufficient to inhibit bacterial growth and/or reapplication may have been necessary sooner - the methods state that a thin layer was applied. As microbial load is one of the foundations of the scoring used in the study, this could affect the final conclusions being drawn.

4. Figure 5 is missing in the file.

Reviewer #2: Well conducted and written study worth publishing, although it is very elaborate and could be abbreviated some. I only have few comments. However, the main problem in understanding is that honey seems to improve pruritus versus placebo improves cytology and clinical signs. Then it is stated that increased cytology scores went along with increased pruritus scores. Makes sense as well, however, then I would assume that honey also should also improve cytology. Did I understand this wrong? This needs to be clearer in the manuscript.

Specific comments:

Line 4: I am not sure if the reference to antibiotic stewardship is needed in the title, may be the Editor should comment on that.

Line 24: "... medical-grade ..."

Line 28: "... the test or the placebo product ..."

Line 37: "... P=0.0443). ..."

Line 40: "... (cytological score ..."

Line 61: A reference should be inserted for the atopic dogs

Line 99: "... inert treatment arm, ..."

Line 102: "... of this study was to ..."

Line 123: "... size was either ..."

Line 172: "... were combined into a ..."

Line 236: "... products was the time ..."

Line 242: "...respectively were H0 ..."

Line 247: "... trial had a maximum ..."

Line 249: "... were blinded, the hypotheses were .... was assumed ..."

Line 254: "... determined ..."

Line 293: "... received the placebo and the other 16 received the medical honey. Of these ..."

Line 296: "... given placebo was discarded ..."

Line 299: "... treated with placebo and 13 with honey ..."

Line 307: "... two with placebo and one with honey ..."

Line 340: "... in the treatment group and of 2 isolates at both times in the placebo group ..."

Line 369: "... additions of breed or sex to the ..."

Line 374: "... for placebo treated dogs than for honey-treated dogs ..."

Line 377 and 378, 408 and 409, 437 and 438, 485 and 486, 488, 498: Ditto

Line 438 and 439: Do the authors mean that the pruritus increased with the cytological score (and thus correlated to the cytology)? But that does not fit the discussion and results. Or that the pruritus decreased with increased cytology? That would not make sense .... Please rephrase.

Line 459-465: Is this needed or could you just delete this?

6. PLOS authors have the option to publish the peer review history of their article (what does this mean?). If published, this will include your full peer review and any attached files.

Reviewer #1: Yes: Nural N Cokcetin

Reviewer #2: No

---

## [Author Response · Author response to Decision Letter 0]

21 Apr 2020

Please find our "response to reviewers" file, where we have addressed all the concerns raised by the Reviewers and Editorial Office. We hope that our answers fulfill their expectations.

---

## [Decision Letter · Decision Letter 1]

28 May 2020

PONE-D-20-01412R1

Medical honey for canine nasal intertrigo: a randomized, blinded, placebo-controlled, adaptive clinical trial to support antimicrobial stewardship in veterinary dermatology

PLOS ONE

Dear Dr. Brosseau,

Thank you for revising and submitting your manuscript to PLOS ONE. I apologise that this review has taken more time than usual - one of the reviewers had difficulties submitting their review which was not picked up and this delayed the process.

After careful consideration, we feel that your manuscript has merit but does not fully meet PLOS ONE’s publication criteria as it currently stands. Therefore, we invite you to submit a revised version of the manuscript that addresses the points raised during the review process.

Both of the original reviewers raise a few issues that do not appear to have been adequately addressed in your resubmission.  Please note their comments and revise the manuscript accordingly. In addition, PLOS ONE assigned an expert in statistical analysis to review your methods and results. This reviewer (#3) has also raised a number of issues that need to be addressed. Overall, many of the comments from the reviewers relate to the fact that the way that the paper is presented (particularly the tables) is quite difficult to understand for non-experts and whatever you can do to improve this will be of help.

We look forward to receiving your revised manuscript.

Kind regards,

Dee A. Carter, PhD

Academic Editor

PLOS ONE

Reviewers' comments:

Reviewer's Responses to Questions

**Comments to the Author**

1. If the authors have adequately addressed your comments raised in a previous round of review and you feel that this manuscript is now acceptable for publication, you may indicate that here to bypass the “Comments to the Author” section, enter your conflict of interest statement in the “Confidential to Editor” section, and submit your "Accept" recommendation.

Reviewer #1: All comments have been addressed

Reviewer #2: (No Response)

Reviewer #3: (No Response)

2. Is the manuscript technically sound, and do the data support the conclusions?

Reviewer #1: Yes

Reviewer #2: Yes

Reviewer #3: Yes

3. Has the statistical analysis been performed appropriately and rigorously? 

Reviewer #1: Yes

Reviewer #2: Yes

Reviewer #3: Yes

4. Have the authors made all data underlying the findings in their manuscript fully available?

Reviewer #1: Yes

Reviewer #2: Yes

Reviewer #3: Yes

5. Is the manuscript presented in an intelligible fashion and written in standard English?

Reviewer #1: Yes

Reviewer #2: Yes

Reviewer #3: Yes

6. Review Comments to the Author

Reviewer #1: The manuscript reads with much more clarity in this revision. Some minor suggestions:

1. Tables are very heavy on statistical analyses and values, some of which are explained in the results text very well and others not. As an example, Table 2 could have a simpler heading (Blinded interim analyses to accept or reject null hypothesis), and the information in the current title could be used in the table legend. Alternatively, it might be useful to state what the hypothesis/null hypothesis is in the table legend for ease of readability. The values in the tables could be reduced to 2 significant figures to simplify the table further. These are just suggestions and not essential to change for the manuscript to be accepted - if it is common in the field to report the results as such, then this suggestion can be overlooked.

When reporting the results here for Table 2, the value in the text refers to -0.370 but this doesn't match values in the table. I had similar confusion with Table 4 - the title could be simplified to what is in the results text (Estimated fixed effects of treatment, time, and treatmend x time on composite cytology score); and then state what is it we are looking at i.e. what values are most important in this table to make your conclusions. It would be helpful to state which value is important and what it is testing - this is done very well for the latter tables in the manuscript (eg. Table 5 where the text specifies the result and conclusion, then reports the Cy_score or trt x time values).

2. Figure legends should specify what panel A and B are (placebo and honey).

3. My comments regarding the sensitivity of the bacterial isolates/species in this study to honey have been addressed and I agree that the microbiological assays do not necessarily have to be conducted. However, I would recommend including the point in the discussion - perhaps in the limitations section. E.g. it would be useful to state that the microbes commonly associated with this condition have been reported as being sensitive to honey (you already have the references showing this in the author comments), and perhaps the antimicrobial effect of honey was not observed here due to the amount used for treatment or frequency of application. It could be a contributor to why the honey was not effective overall since the microbial load is one of the scoring parameters used. There are studies showing that too low a concentration of honey can promote bacterial growth, particularly biofilms that could be cited to make this point.

Reviewer #2: Thank you for addressing most of my previous comments. There are a couple more, but they are fixed easily enough. However, there is still a misleading tenor about the pruritus that is not supported by looking at the dogs from a clinical (and not statistical point of view (see below). Please find my comments below.

Specific comments:

Line 37: “… clinical score (cytological …”

Line 58: “… brachycephalic dogs such as pugs …”

Line 65: “… to preserve their efficacy …”

Line 165: “… An animal health technician randomized each dog’s treatment with a coin toss and kept …”

Line 427: When looking at the dogs’ VAS values in Fig. 4, 4 dogs improved in the honey group and one dog got worse, and four dogs in the placebo group improved and none got worse, even the slope in the dogs improving was similar, and you still state that pruritus improved in the treatment group. (“Fig 4 depicts the effects of treatment on the integer of visual analogue scale values of owner-perceived pruritus in dogs: honey decreased the perceived intensity of pruritus more consistently than placebo.”). I still don’t understand this and even if it is statistically significant, it certainly is clinically absolutely irrelevant in my opinion and the statement that pruritus was decreased more consistently with honey treatment is misleading. This could be clarified in the paragraph starting at line 478, where a statement about the difference between statistically significant and clinically significant could be mentioned, which would completely alleviate my concerns, as the authors have drawn the right conclusions from the study data and the study was well done.

Reviewer #3: I will focus on methods and reporting. Methods are appropriate but unnecessarily complex in my view. Overall, a good manuscript, however, with carefully conducted analyses.

Major

1) There is no clear discussion on power calculations. How did the authors arrived at the final sample? Create a section labelled "power calculations" and expand there. If you didn't perform any, discuss as a major limitation.

2) The analyses are quite complicated because of the repeated design. Why was this chosen? What is the justification. I can understand it when we are dealing with cancer patients and we need an early decision to save lives, but in the context of canine nasal intertrigo seems like overkill.

3) Table 1 on page 12 is not informative for most people so you need to clarify that "actual" is, and the tole of the rejection bounds (explain they are z-values in the text as well).

Minor

1) Double blinded. I am not sure how this applies in this case. I can understand why the dogs were blinded but how were the owners blinded? I would imagine it would be obvious if you were using manuka honey or hydrogel? or am I missing something here?

2) some language corrections are needed.

7. PLOS authors have the option to publish the peer review history of their article (what does this mean?). If published, this will include your full peer review and any attached files.

Reviewer #1: No

Reviewer #2: No

Reviewer #3: No

---

## [Author Response · Author response to Decision Letter 1]

4 Jun 2020

Dear reviewers,

Please find the responses to your concerns included in the composite file.

---

## [Decision Letter · Decision Letter 2]

22 Jun 2020

Medical honey for canine nasal intertrigo: a randomized, blinded, placebo-controlled, adaptive clinical trial to support antimicrobial stewardship in veterinary dermatology

PONE-D-20-01412R2

Dear Dr. Brosseau,

We’re pleased to inform you that your manuscript has been judged scientifically suitable for publication and will be formally accepted for publication once it meets all outstanding technical requirements.

Kind regards,

Dee A. Carter, PhD

Academic Editor

PLOS ONE

Additional Editor Comments (optional):

Reviewers' comments:

Reviewer's Responses to Questions

**Comments to the Author**

1. If the authors have adequately addressed your comments raised in a previous round of review and you feel that this manuscript is now acceptable for publication, you may indicate that here to bypass the “Comments to the Author” section, enter your conflict of interest statement in the “Confidential to Editor” section, and submit your "Accept" recommendation.

Reviewer #3: All comments have been addressed

2. Is the manuscript technically sound, and do the data support the conclusions?

Reviewer #3: Yes

3. Has the statistical analysis been performed appropriately and rigorously? 

Reviewer #3: Yes

4. Have the authors made all data underlying the findings in their manuscript fully available?

Reviewer #3: Yes

5. Is the manuscript presented in an intelligible fashion and written in standard English?

Reviewer #3: Yes

6. Review Comments to the Author

Reviewer #3: I am happy with the author's responses and added text. I found them convincing and clear. I'm not keen on post-hoc power calculations since they are not informative https://www.vims.edu/people/hoenig_jm/pubs/hoenig2.pdf.

7. PLOS authors have the option to publish the peer review history of their article (what does this mean?). If published, this will include your full peer review and any attached files.

Reviewer #3: No

---

## [Editor Report · Acceptance letter]

24 Jul 2020

PONE-D-20-01412R2 

Medical honey for canine nasal intertrigo: a randomized, blinded, placebo-controlled, adaptive clinical trial to support antimicrobial stewardship in veterinary dermatology 

Dear Dr. Brosseau:

I'm pleased to inform you that your manuscript has been deemed suitable for publication in PLOS ONE. Congratulations! Your manuscript is now with our production department. 

Kind regards, 

on behalf of

Prof Dee A. Carter 

Academic Editor

PLOS ONE